# Convergence of Alcohol Consumption and Dietary Quality in US Adults Who Currently Drink Alcohol: An Analysis of Two Core Risk Factors of Liver Disease

**DOI:** 10.3390/nu16223866

**Published:** 2024-11-13

**Authors:** Peng-Sheng Ting, Wei-Ting Lin, Suthat Liangpunsakul, Madeline Novack, Chiung-Kuei Huang, Hui-Yi Lin, Tung-Sung Tseng, Po-Hung Chen

**Affiliations:** 1Division of Gastroenterology and Hepatology, Tulane University School of Medicine, 131 S. Robertson St., New Orleans, LA 70112, USA; pting1@tulane.edu; 2University Medical Center, 2000 Canal St., New Orleans, LA 70112, USA; 3Social, Behavioral, and Population Sciences, Tulane University School of Public Health and Tropical Medicine, 1440 Canal Street, New Orleans, LA 70112, USA; 4Division of Gastroenterology and Hepatology, Department of Medicine, Indiana University School of Medicine, 702 Rotary Circle, Indianapolis, IN 46202, USA; 5Roudebush Veterans Administration Medical Center, 1481 W. 10th Street, Indianapolis, IN 46202, USA; 6Department of Medicine, Tulane University School of Medicine, 1430 Tulane Ave, New Orleans, LA 70112, USA; 7Department of Pathology, Tulane University School of Medicine, 1430 Tulane Ave, New Orleans, LA 70112, USA; 8School of Public Health, Louisiana State University Health Sciences Center, 2020 Gravier Street, New Orleans, LA 70112, USA; 9Division of Gastroenterology and Hepatology, Johns Hopkins University School of Medicine, 1830 East Monument Street, 4th Floor, Baltimore, MD 21287, USA; 10Division of Addiction Medicine, Johns Hopkins University School of Medicine, 1830 East Monument Street, 4th Floor, Baltimore, MD 21287, USA

**Keywords:** National Health and Nutrition Examination Survey, alcoholic beverage type, Healthy Eating Index, diet quality, steatotic liver disease

## Abstract

Background/Objectives: Alcohol consumption and poor dietary habits are on the rise in the United States, posing significant challenges to public health due to their contribution to chronic diseases such as liver failure. While associations between alcohol consumption patterns and diet quality have been explored, the relationship between specific alcoholic beverage types and diet quality remains underexamined. This study aims to compare diet quality among consumers of different alcoholic beverage types. Methods: We conducted a cross-sectional analysis of 1917 current alcohol drinkers from the National Health and Nutrition Examination Survey (NHANES) who completed a 24 h dietary recall survey. Diet quality was assessed using the Healthy Eating Index (HEI), with higher scores indicating superior diet quality. Multivariable logistic regression models were employed to assess differences in HEI between consumers of various alcoholic beverage types, using wine-only drinkers as the reference group and controlling for demographic, socioeconomic, lifestyle, and metabolic syndrome variables. Results: Beer-only drinkers were more likely to have lower income, higher rates of cigarette smoking, and insufficient physical activity compared to other alcohol consumers. In the fully adjusted multivariable model, beer-only drinkers had an HEI score that was 3.12 points lower than wine-only drinkers. In contrast, liquor/cocktail-only and multiple-type drinkers had similar HEI scores to wine-only drinkers. Conclusions: Beer-only consumption is associated with poorer diet quality among alcohol drinkers. Targeted patient education and public health campaigns may be effective in addressing the combined impact of alcohol consumption and poor diet quality on chronic disease risk.

## 1. Introduction

In the United States, a significant portion of the adult population drinks alcohol. In a national survey, 84.1% of adults consumed an alcoholic beverage at least once in their lifetime, and 52.9% had a drink in the past month [1]. This prevalence of alcohol consumption occurs alongside dietary changes characterized by an increasing availability of unhealthy food items [2]. In the United States, this trend toward declining dietary quality is particularly pronounced among the aging population. Between 2001 and 2018, the prevalence of poor diet among older adults increased by 10%, highlighting a concerning trend as the population continues to age [3].

One of the most pressing public health challenges is mitigating the effects of lifestyle factors such as alcohol consumption and poor diet, which are two of the most prevalent risk factors identified in the World Health Organization’s Global Action Plan [4]. The interplay between these factors is multifaceted, but alcohol consumption patterns appear to be related to dietary quality [5,6]. Prior research had demonstrated a dose-dependent worsening of diet quality with increasing alcohol consumption in a cohort that included non-drinkers but did not take into account alcoholic beverage type [6]. To this day, the specific relationship between the type of alcoholic beverage consumed (e.g., wine, beer, liquor) and dietary quality remains a research gap. While it is unlikely that the choice of alcoholic beverage type directly influences dietary quality on a biological level, cultural practices and norms surrounding alcohol consumption (e.g., specific alcoholic beverage and food pairings) can lead to differential associations between beverage type and dietary habits [7].

Regardless of the directionality or causality in the relationship between alcohol consumption patterns and dietary choices, both factors significantly impact the risk of liver disease. Hepatic steatosis is the accumulation of fat in the liver, which can result from metabolic dysfunction and/or alcohol consumption. The rising global prevalence of steatotic liver diseases, including metabolic dysfunction-associated steatotic liver disease (MASLD) and alcohol-related liver disease (ALD), underscores the importance of understanding these lifestyle factors [8,9,10]. Dietary quality alone has been shown to increase the risk of hepatic steatosis [11]. Combined, MASLD and ALD represent a substantial burden, accounting for 37% of a representative cohort of U.S. adults [12]. The gut microbiome, which is influenced by both dietary quality and alcohol consumption, may serve as a key pathophysiological pathway through which these lifestyle factors synergistically inflict liver damage. Understanding the relationship between alcohol and diet quality provides a crucial context for future research aimed at developing targeted interventions for preventing and managing liver disease [13,14].

The type of alcoholic beverage consumed has been linked to a differential risk of liver fibrosis, even among individuals with modest alcohol use in the context of MASLD [15]. Research has also highlighted that wine consumption specifically may offer relative protective effects against liver fibrosis compared to beer and liquor consumption [16,17]. Given the established roles of both alcohol type and dietary quality in influencing liver health, it becomes essential to explore how these two factors interact. Our objective is to investigate the association between the consumption of different types of alcohol and dietary quality.

## 2. Materials and Methods

### 2.1. Study Population

We utilized the pre-pandemic 2017–2020 cycle of the National Health and Nutrition Examination Survey (NHANES), which provides a comprehensive assessment of the health and nutritional status of adults in the United States [18]. NHANES employs a combination of in-depth interviews and physical examinations conducted on a nationally representative sample of U.S. adults. NHANES data collection occurs biennially under the oversight of the National Center for Health Statistics, a branch of the Centers for Disease Control and Prevention. Prior to data collection, informed consent was obtained from each study participant, and the research protocol received approval from the National Center for Health Statistics Research Ethics Review Board.

We included participants who were ≥20 years old, identified as current alcohol drinkers by the 12-month Alcohol Use Questionnaire (ALQ), and consumed alcohol within a completed 24 h dietary recall. We defined a drink as a standard drink containing 14 g of pure alcohol (approximately 12 oz. beer, 5 oz. wine, or 1.5 oz. liquor), following the guidelines of the National Institute on Alcohol Abuse and Alcoholism (NIAAA) [19]. Using the “What We Eat in America” Food Categories in the 24 h dietary recall survey, we classified alcohol drinkers into 4 groups: beer-only, wine-only, liquor/cocktail-only, and multiple types (if consumed ≥2 types of alcoholic beverages) [20].

### 2.2. Healthy Eating Index

Dietary intake data were collected from two-day, 24 h dietary recall interviews using the U.S. Department of Agriculture Food Patterns Equivalence Database (FPED). The FPED dataset, along with the 24 h dietary interview files (DR1TOT and DR2TOT) recorded in NHANES, was utilized to estimate the Healthy Eating Index-2015 (HEI). HEI is a widely recognized measure of dietary quality recommended by the Dietary Guidelines for Americans [21]. The HEI consists of 13 components that assess various aspects of dietary quality, divided into four moderation components and nine adequacy components. The four moderation components—sodium, refined grains, added sugars, and saturated fats—are designed to be limited in a healthy diet and are each scored from 0 to 10, with lower intakes receiving higher scores. The adequacy components, which include whole grains, dairy, and fatty acids (scored from 0 to 10), as well as total fruits, whole fruits, total vegetables, greens and beans, total protein foods, seafood, and plant proteins (each scored from 0 to 5), evaluate the intake of food groups and nutrients that should be consumed in adequate amounts to support health. The sum of scores for all 13 components produces the total HEI, which ranges from 0 to 100. A higher HEI score reflects a healthier diet and has been associated with a lower risk of developing MASLD [11].

### 2.3. Total Nutrient Intake

The nutrient composition of participants’ daily diets was derived from DR1TOT and DR2TOT, which included total energy, sugar, alcohol, protein, carbohydrates, total fat, saturated fatty acids, monounsaturated fatty acids, and polyunsaturated fatty acids. Detailed guides and protocols for the dietary recall interviews, including the methods used to measure and record nutrient intake, are available on the NHANES website [22].

### 2.4. Covariates

Sociodemographic factors were collected using the Computer-Assisted Personal Interview (CAPI) system conducted by trained interviewers. These factors included age, gender, race, and the poverty income ratio (PIR). PIR is defined as the total household income divided by the poverty threshold for household size as defined by the US Census Bureau [23]. PIR serves as a standardized measure of socioeconomic status that allows for comparison of individual income across geography and time within the large area of the United States and is classified into 5 categories of increasing income: <1.00 (below poverty), 1.00–1.99, 2.00–2.99, 3.00–3.99, and ≥4 [24].

Lifestyle patterns and comorbidities were also gathered via the CAPI system. We identified current heavy alcohol drinkers (by NIAAA definitions), a group at increased risk for alcohol-related health issues [19]. For men, heavy drinking is defined as consuming >14 drinks/week or >4 drinks on a typical drinking day; for women, it is consuming >7 drinks/week or >3 drinks on a typical drinking day. We calculated the quantity of alcohol consumption per week using two questions within the NHANES Alcohol Use Questionnaire (ALQ) to identify heavy drinking: ALQ121: “How often did you drink any type of alcoholic beverage during the past 12 months?” and ALQ130: “On average, how many drinks did you have on those days you drank alcoholic beverages?” We used ALQ151 to further evaluate the participants’ drinking history; the ALQ151 identifies individuals with any history of sustained heavy alcohol use in any period in a participant’s life. Participants who responded positively to ALQ151 but did not report current heavy drinking were characterized as having a remote history of heavy drinking.

Sufficient physical activity was defined in accordance with the WHO recommendations for adults [25]. Medical conditions were identified using the Medical Conditions Questionnaire (MCQ), which includes a wide range of pre-existing medical conditions such as hypertension, congestive heart failure, heart attack, and chronic lung disease.

Metabolic syndrome and body mass index (BMI) were assessed using laboratory datasets. Metabolic syndrome was defined as having 3 out of 5 abnormal metabolic syndrome components as defined by the Adult Treatment Panel III [26].

### 2.5. Statistical Analysis

Categorical variables were reported as percentages, and continuous variables were expressed as mean ± standard error. We utilized chi-square analysis for categorical data and analysis of variance for continuous data. To examine the association between types of alcoholic beverage intake and HEI, we conducted multiple linear regression analyses. These models were designed to account for confounders and isolate the relationship between alcoholic beverage type and dietary quality. The regression models were structured as follows:

Model 1: Adjusted for sociodemographic factors, including age, gender, race, and the ratio of family income to poverty (PIR). This initial model accounted for basic demographic characteristics that could influence both alcohol consumption patterns and dietary quality.

Model 2: Adjusted for all covariates in Model 1, with additional adjustments for lifestyle factors and comorbidities. These included smoking status, presence of heavy alcohol use, physical activity levels, and medical conditions identified through the Medical Conditions Questionnaire (MCQ). This model aimed to control for behaviors and health conditions that could impact dietary quality and alcohol consumption independently of sociodemographic factors.

Model 3: Adjusted for all covariates in Model 2, with further adjustments for the presence of metabolic syndrome. This final model considered the combined effects of metabolic health and lifestyle factors on the HEI score, providing a more comprehensive assessment of how metabolic syndrome might interact with alcohol consumption and dietary quality.

Covariables were selected based on the change-in-estimate method and considerations of previously identified confounders [27,28]. To account for the complex survey design of NHANES, we utilized survey modules in our statistical analyses. These modules adjust for the unequal sampling probabilities and ensure that estimates are representative of the national population. All data management and statistical analyses were performed using Stata software (StataCorp, College Station, TX, USA), version 17.

We conducted a sensitivity analysis excluding individuals with a remote history of heavy alcohol drinking (*n* = 152) from the multivariable regression models. By removing participants with a history of heavy drinking, we sought to minimize the potential confounding effects of longstanding alcohol-related dietary habits and health consequences, which could differ from those associated with more recent alcohol use.

## 3. Results

### 3.1. The Detailed Distribution of Demographics, Lifestyle Patterns, and Comorbidities Among U.S. Adults Who Consume Different Types of Alcoholic Beverages

A total of 1917 current alcohol drinkers were included in our study. The distribution of alcoholic beverage type consumption was as follows: 38.9% beer-only, 21.8% wine-only, 18.2% liquor/cocktail-only, and 21.0% multiple-type. Wine-only consumers were relatively older and predominantly women, with an average age of 53.4 years (*p* < 0.001) and the lowest proportion of men at 27.6% (*p* < 0.001). Alcoholic beverage choice varied significantly across different racial groups (*p* = 0.002). Socioeconomic differences were evident, as beer-only drinkers had the highest proportion living below the poverty line at 12.7% and the lowest proportion with a PIR ≥4 (*p* < 0.001); this suggests beer-only drinkers had the lowest income levels among all drinkers. Behavioral patterns also differed in beer-only drinkers, who had the highest prevalence of current cigarette use at 27.8% (*p* ≤ 0.001) and the lowest prevalence of sufficient physical activity at 42.2% (*p* = 0.002). Conversely, exclusive liquor or cocktail drinkers exhibited the highest prevalence of obesity, as measured by both waist circumference and BMI. Regarding alcohol consumption history, beer-only drinkers had the highest prevalence of a remote history of heavy drinking at 10%. In contrast, wine-only drinkers had the lowest prevalence of both current and remote heavy drinking at 14.3% and 0.9%, respectively. Table 1 presents the sociodemographics, lifestyle patterns, and comorbidities of our study population by alcoholic beverage type consumed.

### 3.2. Daily Macronutrient Intake by the Type of Alcoholic Beverage Consumed

Analysis of macronutrient composition among different types of alcoholic beverage consumers revealed distinct patterns in daily caloric and nutrient intake. Adjusted for body weight, beer-only drinkers had the highest total daily caloric consumption at 29.6 kcal/kg, compared to 28.8 kcal/kg in the overall sample (*p* = 0.047). Beer-only consumers also had the highest carbohydrate intake, averaging 258 g (*p* < 0.001), and the highest saturated fatty acids intake, at 31.7 g (*p* = 0.005). In terms of total daily sugar intake, the liquor/cocktail-only group reported the highest amount at 102.9 g. This was slightly higher than the 101.1 g in beer-only drinkers (*p* = 0.007). Notably, sugar intake from alcohol was significantly higher in liquor/cocktail-only drinkers (12.8 g) compared to beer-only drinkers (1.8 g), indicating beer-only drinkers have alternative non-alcohol sources of excess sugars (see Appendix A). The highest average daily alcohol consumption was observed among multiple-type drinkers, averaging 46.9 g, significantly higher than the 31.3 g in the overall sample (*p* < 0.001). Table 2 provides a detailed breakdown of daily macronutrient intake by the alcoholic beverage type consumed.

### 3.3. Comparative Dietary Quality Across Alcohol Beverage Types

In the assessment of dietary quality standardized by HEI, beer-only drinkers had the lowest total HEI score, averaging 49.3, compared to 51.9 for the overall sample (*p* = 0.004). Beer-only drinkers also had the lowest scores on several individual HEI components, including refined grains, total vegetables, greens and beans, total fruits, seafood and plant proteins, and fatty acids (Ps ≤ 0.020). The unadjusted HEI score was 5.8 points lower in beer-only consumers compared with wine-only. Exclusive liquor and cocktail drinkers had the lowest scores on the added sugar component, which aligns with their high total daily sugar intake. In contrast, wine-only drinkers exhibited the highest total HEI score of 55.1 and scored the highest on several individual HEI components, including added sugar, total vegetables, greens and beans, total fruits, whole fruits, whole grains, and seafood and plant proteins (Ps ≤ 0.020). Table 3 presents a comprehensive overview of the HEI components and total HEI scores by the alcoholic beverage type consumed.

### 3.4. Exploring the Relationship Between Alcohol Consumption and Dietary Quality: Results from Multivariable Regression Analysis

We conducted multivariable regression analyses in our study population of alcohol drinkers using wine-only drinkers as the reference group. In Model 1, which was adjusted for sociodemographic factors, beer-only drinkers exhibited a lower HEI score of −4.47 (95% CI: −6.84, −2.09) compared to wine-only drinkers. This represents a smaller difference than the unadjusted HEI difference of −5.80. Model 2, which further adjusted for lifestyle factors and medical conditions, the disparity in HEI between beer-only and wine-only drinkers persisted, with a lower HEI of −3.14 (95% CI: −5.50, −0.78) observed among beer-only drinkers. In the fully adjusted Model 3, which additionally accounted for the presence of metabolic syndrome, beer-only drinkers continued to show a lower HEI by −3.12 (95% CI: −5.50, −0.74) (see Figure 1). Sensitivity analyses, excluding remote heavy drinkers (*n* = 152), revealed a strengthened association where beer-only drinkers had a significantly lower HEI of −3.97 (95% CI: −6.28, −1.66) in the fully adjusted Model 3 (see Appendix A). Importantly, liquor/cocktail-only and multiple-type alcoholic beverage drinkers had HEI scores similar to those of wine-only drinkers across all models.

## 4. Discussion

In our study, we observed notable demographic and lifestyle differences among beer drinkers compared to consumers of other types of alcoholic beverages. We found beer-only drinkers had a higher prevalence of men, lower income levels, current cigarette smoking, and insufficient physical activity. To our knowledge, this is the first study to report an association between the HEI and the type of alcoholic beverage consumed. We found that among all current alcohol consumers, beer-only consumption was associated with the lowest relative dietary quality even after adjusting for previously described confounders such as sociodemographic and lifestyle differences.

### 4.1. Historical Comparisons of Dietary Quality as a Function of Alcoholic Beverage Choice

The relationship between the choice of alcoholic beverage type and dietary quality has been explored, but studies vary widely in design and outcomes. For instance, various studies have assessed simple proportions of fast-food consumption, budget expenditure on healthy foods, or the intake of specific macronutrients. Only one study has directly measured the HEI, but it did not find a significant impact of alcoholic beverage type on HEI scores [7,29]. Our findings align with those of McCann et al., who reported that beer-only drinkers had the lowest intake of fruits and juices among alcohol consumers. In particular, male beer drinkers had the lowest dairy intake, while female beer drinkers had the lowest grain intake [30]. Similarly, Barefoot et al. found that individuals with a wine preference reported higher intakes of fiber, fruits, and vegetables and lower intakes of cholesterol, saturated fats, red meats, and fried meats [28].

One hypothesis for the association we found is that the context of alcohol consumption may influence dietary behavior and vice versa. For instance, a full-course meal that contains fruits and vegetables with high diet quality often accompanies wine or spirit-based drinks, and high sodium or fried foods with low diet quality may demand carbonated alcoholic beverages such as beer. This is an area that remains uninvestigated, but a descriptive study set in Great Britain reported wine is more often paired with a formal meal (45.8%, vs. 25.6% for beer), whereas beer is more often paired with a snack (33.2%, vs. 25.6% for wine) [31]. However, due to the limitation of the 24 h dietary recall instrument, we could not localize the alcohol consumption to each specific meal.

### 4.2. Impact of Dietary Quality on Liver Disease

Our observation of lower diet quality among beer-only drinkers, as indicated by HEI scores, is especially significant given the established link between poor dietary quality and an increased risk of MASLD [11,32]. Zhang et al. demonstrated that individuals in the highest quartile of HEI had a 43.3% reduced risk of hepatic steatosis compared to those in the lowest quartile [11]. Furthermore, Huang et al. found that higher dietary quality was associated with decreased all-cause and cardiovascular-related mortality, particularly among those with low physical activity [33]. This is especially relevant to our study because beer-only drinkers exhibited the lowest rates of physical activity, which could compound the adverse effects of poor dietary quality. Taken together, these findings suggest for individuals with poor dietary quality, such as beer-only drinkers, increasing physical activity might offer a modifiable strategy to enhance overall health and potentially offset some of the negative impacts associated with a poor diet.

### 4.3. Associations of Alcohol Consumption Pattern with Dietary Quality

Our analysis of current alcohol consumers revealed that diet quality did not differ significantly between current heavy drinkers and light drinkers (see Appendix A). This finding contrasts with previous research that reported a decline in dietary quality with increasing alcohol consumption. For instance, Breslow et al. documented a decrease in HEI with higher alcohol consumption, showing a drop from 55.9 among men drinking <1 drink/day to 41.5 among those consuming ≥5 drinks/day and from 59.5 among women drinking <1 drink/day to 51.8 among those drinking ≥3 drinks/day [6]. A separate study found a decrease in HEI with decreasing alcohol consumption frequency and increasing alcohol quantity on each occasion, demonstrating the poorest diet quality among those consuming the highest quantity of alcohol per day at the lowest frequency in a stratified analysis [5]. These findings imply that binge drinking behavior, characterized by consuming large amounts of alcohol in a single session, is associated with decreased dietary quality. It is noteworthy that Breslow et al. did not analyze diet quality by the type of alcoholic beverage due to sample size limitations, leaving a gap in understanding how different types of alcohol might interact with diet quality. In contrast, our study exclusively included participants who consumed alcohol in the past year and assessed both the types of alcoholic beverages consumed and diet quality through the same two 24 h dietary recalls for internal consistency.

To further investigate the role of remote heavy alcohol use in dietary quality, we conducted a sensitivity analysis excluding current light alcohol drinkers with a history of heavy alcohol use. This analysis revealed a stronger association between poorer diet quality and exclusive beer consumption compared to the overall sample. Breslow et al. provided insight into this dynamic by comparing dietary quality on drinking versus non-drinking days, which found worse diet quality on drinking days [34]. This suggests that the link between alcohol consumption and diet quality may not only be influenced by long-term patterns but also by short-term behaviors and consumption habits.

### 4.4. Impact of Alcohol and Diet Quality on Liver Fibrosis

The association between alcoholic beverage type and diet quality is significant due to its potential implications for liver disease risk. Our findings show that wine drinkers report higher diet quality compared to beer drinkers, which may help explain the previously observed mitigation of liver fibrosis risk associated with a higher proportion of wine in total alcohol consumption. In a study by Becker et al., an increased proportion of wine consumption was linked to a decreased risk of liver fibrosis [16]. This is supported by Mitchell et al., who found that in an Australian cohort of predominantly light to moderate alcohol consumers with MASLD, exclusive wine drinkers had a lower risk of advanced liver fibrosis compared to exclusive beer drinkers [17]. Similarly, Becker et al. demonstrated in a Danish cohort that a higher proportion of wine in total alcohol consumption was associated with a reduced risk of developing cirrhosis, with those consuming more than 50% of their alcohol as wine having a relative risk of 0.3 (95% CI: 0.2–0.5) for cirrhosis. While increasing alcohol consumption generally raises the risk of cirrhosis regardless of the type of alcoholic beverage, the protective effect associated with wine was not observed in beer drinkers at any consumption level [16]. These findings underscore the potential influence of dietary quality and alcoholic beverage type on liver health. Wine, associated with a higher HEI, appears to be linked with better dietary quality and a potentially lower risk of liver-related complications compared to beer.

Although we previously found a higher risk of liver fibrosis among liquor/cocktail-only drinkers with modest alcohol use compared to consumers of other alcoholic beverage types, we did not find a worse diet quality among liquor/cocktail-only drinkers that may have explained a higher fibrosis risk among this group [15]. It is worth noting that the population in this study differs in that we did not exclude heavy drinkers, limiting lateral comparison.

### 4.5. Changes in Diet Quality and Alcohol Consumption Pattern with Age

Time is a crucial, often overlooked, dimension in the discussion of alcohol consumption and its relationship with diet. Parekh et al. highlighted a notable trend in the Framingham Heart Study Offspring Cohort, showing a general decline in alcohol consumption with age over 37 years of follow-up. This study also noted a shift in beverage preference, with increased wine consumption (from 1.4 to 2.2 drinks/week) and decreased consumption of beer (from 3.4 to 1.5 drinks/week) and cocktails (from 2.8 to 1.2 drinks/week) [35]. Our findings corroborate this trend, as we observed that wine drinkers are relatively older compared to other drinkers. This temporal shift in alcohol consumption patterns may be linked to changes in dietary habits and overall diet quality as people age. As Parekh et al. eloquently acknowledged, “diet quality may vary among wine, spirit, and beer drinkers, but the influence of beverage preference on the alcohol-diet link has not been elucidated” [35]. Our study contributes to this conversation by demonstrating that exclusive beer drinkers have a lower diet quality compared to wine and spirits drinkers. However, understanding how beverage preference and diet quality interact and change over time requires further investigation.

### 4.6. Strengths and Limitations

Our study has several strengths. Firstly, it is a nationally representative sample of US adults, enhancing the generalizability of our findings. Secondly, we utilized duplicate 24 h dietary recall methods to quantify/categorize alcohol and standardize diet quality with HEI, which is more reliable than dietary records for quantifying energy intake and alcohol consumption [36]. Thirdly, the simultaneous recording of alcoholic beverage type and dietary quality during the same instrument ensures internal consistency. Moreover, we adjusted for the previously observed association between HEI and the amount of alcohol consumption in addition to the type of alcoholic beverage consumed, allowing for a more nuanced understanding of the relationship between alcohol type and diet quality [6].

Despite these strengths, our study has limitations. The cross-sectional design restricts our ability to infer causation. The COVID-19 pandemic occurred after the study period, and we recognize both dietary and alcohol behaviors may have changed after the pandemic in uncertain ways that our study was not designed to detect. There is also the potential for residual or unmeasured confounding, although we have made efforts to adjust for the most pertinent factors related to both alcohol and diet. Additionally, NHANES sampling may disproportionately exclude unhoused individuals, potentially underrepresenting ethnic minorities and those with lower socioeconomic status. Finally, we did not consider the potential cultural impact of diet and alcohol behaviors because of NHANES’ limitations in identifying cultural concepts, such as pairing specific alcoholic beverage types to food in the same meal. Future prospective studies collecting data on cultural context would be beneficial in elucidating the influence of social/cultural norms on both alcohol use and dietary quality.

## 5. Conclusions

In summary, among US adult alcohol consumers, beer consumption is associated with lower relative dietary quality compared to wine and spirit consumption. The association persists even after adjusting for relevant variables, including socioeconomic status, a commonly cited factor influencing dietary differences related to alcoholic beverage choice. Future prospective studies that follow individuals over time could help identify determinants of alcoholic beverage preferences and better understand how these preferences interact with diet quality. Additionally, such research should explore the combined effects of alcoholic beverage type and diet on the development of steatotic liver disease.

## Figures and Tables

**Figure 1 nutrients-16-03866-f001:**
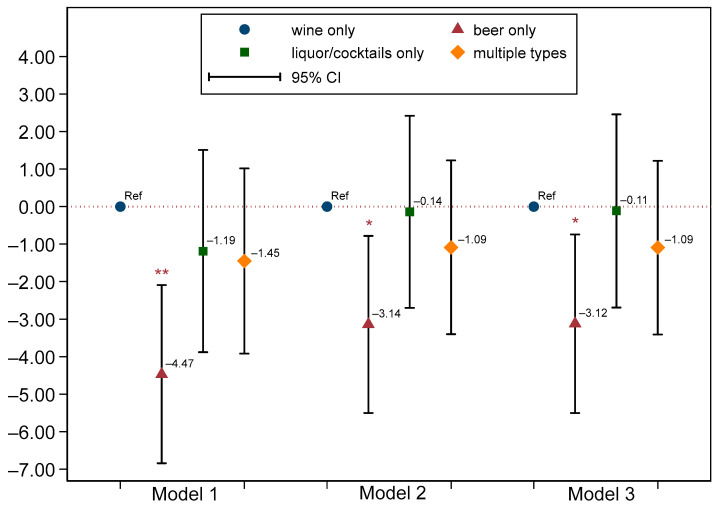
Associations between type of alcoholic beverage and Healthy Eating Index scores among all current alcohol drinkers. Model 1 was adjusted for age, sex, race, and poverty-to-income ratio. Model 2 was adjusted for covariates in Model 1 and the status of cigarettes and amount of alcohol use, physical activity, and medical conditions. Model 3 was adjusted for covariates in Model 2 and the presence of metabolic syndrome. *, *p* < 0.05; **, *p* < 0.01 when compared to wine only. Abbreviation, Ref, reference group.

**Table 1 nutrients-16-03866-t001:** The demographics, lifestyle patterns, and comorbidities among different types of alcohol drinkers.

		Type of Alcoholic Beverage Consumed				
	Total	Beer Only	Wine Only	Liquor/Cocktails Only	Multiple Types	*p* Value
Sample size ^1^ (survey-weighted % ^2^)	1917 (100)	798 (38.9)	394 (21.8)	379 (18.2)	346 (21.0)	
Demographic factors						
Age (years), mean ± standard error	46.8 ± 0.8	44.8 ± 1.1	53.4 ± 1.2	46.1 ± 1.2	44.2 ± 1.6	<0.001
Male gender	56.5%	73.4%	27.6%	48.8%	62.1%	<0.001
Race						
non-Hispanic White	67.7%	64.4%	72.6%	61.9%	74.0%	0.002
non-Hispanic Black	10.4%	10.3%	9.2%	13.2%	9.5%	
Mexican American	7.4%	11.6%	2.7%	7.5%	4.3%	
Other Hispanic	7.1%	7.3%	7.2%	9.8%	4.1%	
Others	7.3%	6.3%	8.3%	7.6%	8.1%	
Poverty to income ratio						
<1.00 (Below poverty)	9.2%	12.7%	5.7%	9.6%	6.0%	<0.001
1.00–1.99	13.7%	19.0%	10.2%	15.8%	5.4%	
2.00–2.99	13.8%	17.3%	9.0%	11.4%	14.1%	
3.00–3.99	12.5%	10.3%	11.0%	15.4%	15.8%	
≥4	50.8%	40.7%	64.0%	47.8%	58.7%	
Lifestyle patterns						
Current cigarettes smoker	19.9%	27.8%	10.0%	19.9%	15.3%	<0.001
Amount of alcohol use						
current light drinking ^3^	72.9%	68.8%	84.8%	72.4%	68.3%	0.001
current heavy drinking	20.9%	21.2%	14.3%	19.8%	27.9%	
remote history of heavy drinking ^4^	6.3%	10.0%	0.9%	7.8%	3.8%	
Sufficient physical activity ^5^	50.3%	42.2%	59.8%	46.1%	59.3%	0.002
Presence of medical conditions ^6^	58.1%	57.5%	62.5%	58.7%	54.3%	0.574
Clinical examination						
Metabolic syndrome components						
Large waist circumference ^7^	53.2%	51.8%	50.7%	63.2%	49.7%	0.039
High blood pressure ^8^	43.2%	46.0%	43.9%	47.1%	33.5%	0.168
Elevated triglycerides ^9^	30.2%	35.3%	24.4%	28.9%	28.1%	0.251
Low HDL-C ^10^	16.5%	18.8%	16.7%	19.7%	9.3%	0.142
Elevated glucose ^11^	9.4%	10.4%	10.2%	11.1%	5.0%	0.090
Metabolic syndrome	23.8%	27.7%	21.4%	27.3%	16.1%	0.045
BMI						
Normal weight (<25)	29.6%	25.3%	40.1%	21.2%	33.6%	0.031
Overweight (25 to 29.9)	33.7%	36.0%	35.7%	31.2%	29.8%	
Obese (≥30)	36.7%	38.7%	24.2%	47.7%	36.6%	

^1^ Raw number of participants in this study without adjusting for sample survey design. ^2^ Results were obtained after adjusting for sample weights and complex study design. ^3^ Current light drinking is defined as without past history of heavy drinking by ALQ151. ^4^ Remote history of heavy drinking is defined as current light drinking but positive ALQ151. ^5^ Sufficient activity is defined as ≥150 min of moderate-intensity or ≥75 min of vigorous-intensity exercise during leisure time per week. ^6^ Medical conditions include asthma, diabetes, COPD, arthritis, hypertension, congestive heart failure, heart attack, weak/failing kidneys, liver conditions, angina, thyroid problems, and cancer/malignancy. ^7^ Large waist circumference is defined as ≥102 cm in men and ≥88 cm in women. ^8^ High blood pressure is defined as systolic blood pressure ≥130 mm Hg or diastolic blood pressure ≥85 mm Hg. ^9^ Elevated triglycerides are defined as ≥150 mg/dL or current use of medication for elevated triglycerides. ^10^ Low high-density lipoprotein cholesterol is defined as HDL-C, <40 mg/dL in men and <50 mg/dL in women. ^11^ Elevated blood glucose is defined as fasting glucose ≥100 mg/dL or current use of antidiabetic medication.

**Table 2 nutrients-16-03866-t002:** Daily total and macronutrients intake of different types of alcoholic beverage drinkers.

		Type of Alcoholic Beverage Consumed				
	Total	Beer Only	Wine Only	Liquor/Cocktails Only	Multiple Types	*p* Value
Sample size ^1^ (survey-weighted % ^2^)	1917 (100)	798 (38.9)	394 (21.8)	379 (18.2)	346 (21.0)	
Daily dietary intake						
Total energy (kcal), mean ± SE	2314 ± 33	2464 ± 38	1980 ± 52	2342 ± 86	2358 ± 85	<0.001
Total energy intake per kg (kcal/kg), mean ± SE	28.8 ± 0.5	29.6 ± 0.6	26.9 ± 0.7	28.6 ± 1.4	29.3 ± 1.0	0.047
Sugar (gram), mean ± SE	94.9 ± 2.3	101.1 ± 4.2	83.9 ± 3.6	102.9 ± 6.7	87.8 ± 3.7	0.007
Fat (gram), mean ± SE	91.6 ± 1.5	97.0 ± 1.7	83.1 ± 2.8	92.7 ± 4.0	89.3 ± 3.6	0.077
Alcohol (gram), mean ± SE	31.3 ± 1.0	28.8 ± 1.8	18.4 ± 1.1	34.3 ± 3.8	46.9 ± 3.2	<0.001
Protein (gram), mean ± SE	86.5 ± 1.2	91.8 ± 2.0	75.9 ± 2.3	89.8 ± 3.1	85.0 ± 3.4	0.001
Protein intake per kg (gram/kg), mean ± SE	1.1 ± 0.02	1.1 ± 0.03	1.0 ± 0.03	1.1 ± 0.05	1.1 ± 0.04	0.575
Carbohydrate (gram), mean ± SE	235 ± 4.0	258 ± 6.6	205 ± 6.1	233 ± 12.2	227 ± 8.1	<0.001
Saturated fatty acid (gram), mean ± SE	29.4 ± 0.5	31.7 ± 0.6	26.7 ± 1.0	29.5 ± 1.3	27.7 ± 1.4	0.005
Monounsaturated fatty acid (gram), mean ± SE	31.6 ± 0.6	33.2 ± 0.7	28.6 ± 0.9	32.2 ± 1.7	31.0 ± 1.4	0.009
Polyunsaturated fatty acid (gram), mean ± SE	21.3 ± 0.3	31.9 ± 0.5	19.3 ± 0.8	21.4 ± 1.3	21.9 ± 0.7	0.120

^1^ Raw number of participants in this study without adjusted for sample survey design. ^2^ Results were obtained after adjusting for sample weights and complex study design. Abbreviations: SE, standard error.

**Table 3 nutrients-16-03866-t003:** Healthy Eating Index (HEI) score and individual components among different types of alcohol drinkers.

		Type of Alcoholic Beverage Consumed				
	Total	Beer Only	Wine Only	Liquor/Cocktails Only	Multiple Types	*p* Value
Sample size ^1^ (survey-weighted % ^2^)	1917 (100)	798 (38.9)	394 (21.8)	379 (18.2)	346 (21.0)	
Dietary components for HEI-2015 score						
Moderation components						
Sodium (0–10), mean ± SE	4.9 ± 0.1	4.9 ± 0.2	4.7 ± 0.2	4.9 ± 0.2	5.2 ± 0.3	0.564
Refined grains (0–10), mean ± SE	6.9 ± 0.1	6.5 ± 0.1	7.0 ± 0.2	7.4 ± 0.2	6.8 ± 0.3	0.011
Saturated fats (0–10), mean ± SE	5.7 ± 0.1	5.4 ± 0.2	5.4 ± 0.2	5.7 ± 0.2	6.6 ± 0.3	0.005
Added sugar (0–10), mean ± SE	7.9 ± 0.1	7.7 ± 0.2	8.2 ± 0.2	7.6 ± 0.2	8.1 ± 0.2	0.011
Adequacy components						
Total vegetables (0–5), mean ± SE	3.0 ± 0.1	2.9 ± 0.1	3.3 ± 0.1	3.0 ± 0.1	3.0 ± 0.2	0.020
Greens and beans (0–5), mean ± SE	1.6 ± 0.1	1.3 ± 0.1	2.0 ± 0.1	2.0 ± 0.2	1.7 ± 0.2	0.004
Total fruits (0–5), mean ± SE	1.7 ± 0.1	1.5 ± 0.1	2.2 ± 0.1	1.8 ± 0.1	1.5 ± 0.1	<0.001
Whole fruits (0–5), mean ± SE	1.8 ± 0.1	1.6 ± 0.1	2.4 ± 0.1	1.7 ± 0.2	1.5 ± 0.2	<0.001
Whole grains (0–10), mean ± SE	2.2 ± 0.1	2.0 ± 0.1	3.0 ± 0.3	2.1 ± 0.2	1.7 ± 0.2	0.003
Total dairy (0–10), mean ± SE	4.4 ± 0.1	4.5 ± 0.1	4.5 ± 0.2	4.8 ± 0.2	3.8 ± 0.3	0.008
Total protein foods (0–5), mean ± SE	4.3 ± 0.04	4.2 ± 0.1	4.4 ± 0.1	4.4 ± 0.1	4.3 ± 0.1	0.184
Seafood and plant proteins (0–5), mean ± SE	2.4 ± 0.1	2.0 ± 0.1	2.9 ± 0.2	2.5 ± 0.2	2.5 ± 0.2	<0.001
Fatty acids (0–10), mean ± SE	5.1 ± 0.1	4.7 ± 0.1	5.1 ± 0.2	5.0 ± 0.2	5.8 ± 0.2	0.006
Total HEI-2015 score	51.9 ± 0.6	49.3 ± 0.7	55.1 ± 0.8	52.8 ± 1.1	52.6 ± 0.9	0.004

^1^ Raw number of participants in this study without adjustment for sample survey design. ^2^ Results were obtained after adjusting for sample weights and complex study design. Abbreviations: HEI, healthy eating index; SE, standard error.

## Data Availability

NHANES data are publicly available at https://wwwn.cdc.gov/nchs/nhanes/continuousnhanes/default.aspx?Cycle=2017-2020 (accessed on 1 September 2024).

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
