# Peer review of "Convergence of Alcohol Consumption and Dietary Quality in US Adults Who Currently Drink Alcohol: An Analysis of Two Core Risk Factors of Liver Disease"

_nutrients, 2024, doi:10.3390/nu16223866_

Round 1
Reviewer 1 Report
Comments and Suggestions for Authors
By their study, the authors aim at comparing alcohol consumers (1917 subjects: bear-only, wine-only, liquor/cocktail-only drinkers) and their dietary quality. By a cross-sectional analysis, the authors reach the conclusion that beer-only drinkers are associated with a poor quality diet as well as lower income, higher rates of cigarette smoking, and insufficient physical activity compared to other alcohol consumers. In contrast, liquor/cocktail-only and multiple type drinkers had HEI scores like wine-only drinkers.
The study is interesting and well conducted, however the criticisms raised by this reviewer are as follows:
1) In the manuscript surely of interest but long and of complex/difficult to understand, the authors are strongly suggested to show the results by graphics and/or histograms instead of tedious and devoid of visual impact Tables.
2) The study population is made by non-Hispanic White (67.7%), non-Hispanic Black (10.4%) and other belonging to different ethnic groups. It shows up that the White group is much larger than the others. The question is: which was the criteria adopted to define these different proportions? The authors wanted to propose a picture of the USA society or what else or nothing at all? It is well-known that different ethnic groups have quite different customs deriving from different culture and socio-economic possibilities.
However, Table 1 shows only the percent of the Total. How would it be if the authors would have shown the percent of each population? Quite different of course, but the problem is that the authors choose to mainly represent non-Hispanic white. In this reviewer’s opinion if the authors would be aware to represent a particular situation, picture of the society, they must well define this point in the text. Otherwise, the chosen population set is strongly inhomogeneous without an underlying rationale. In any case, the percent of each and the different data obtained must be shown and discussed.
3) The study completely lacks investigation on the cultural level of the people: instruction level, cultural habits that play a crucial importance in defining drinking and eating habits too. This point must be added, and its weight well specified in the study.
4) The poverty to income ratio must be well specified in the text. This reviewer did not well understand the meaning of the numbers of the different groups, nor how these groups were decided and adopted.
5) No other exclusion criteria than alcohol abuse? It seems very narrow to limit and define the investigated set.
6) The authors defined an only-bear and an only-wine groups. Indeed, USA is a country where the daily use of wine is not present in the population habits as it would be for Mediterranean (Spanish, Italian, French) population. The reason is a cultural, agricultural and hence economic weight. Which the reason why the authors choose only-wine as one of the groups for their study? In this reviewer’s opinion this choice is not as obvious as it would be in Italy or French. This point would have been defined and discussed in the manuscript.
7) Based on the above observation, would the authors think possible to apply their study to other countries different from USA and what do the authors expect?
8) The limitations of the study are vague and difficult to understand in their practical application.
Comments on the Quality of English Languageminor editing requested
Author Response
Comment 1: In the manuscript surely of interest but long and of complex/difficult to understand, the authors are strongly suggested to show the results by graphics and/or histograms instead of tedious and devoid of visual impact Tables.
Response 1: We appreciate the comment, we have streamlined language to make the manuscript more concise along with edits to reflect your comments. We also replaced our main results Table 4 with Figure 1 to improve demonstration of findings with graphics
Comment 2: The study population is made by non-Hispanic White (67.7%), non-Hispanic Black (10.4%) and other belonging to different ethnic groups. It shows up that the White group is much larger than the others. The question is: which was the criteria adopted to define these different proportions? The authors wanted to propose a picture of the USA society or what else or nothing at all? It is well-known that different ethnic groups have quite different customs deriving from different culture and socio-economic possibilities.
However, Table 1 shows only the percent of the Total. How would it be if the authors would have shown the percent of each population? Quite different of course, but the problem is that the authors choose to mainly represent non-Hispanic white. In this reviewer’s opinion if the authors would be aware to represent a particular situation, picture of the society, they must well define this point in the text. Otherwise, the chosen population set is strongly inhomogeneous without an underlying rationale. In any case, the percent of each and the different data obtained must be shown and discussed.
Response 2: We utilized NHANES dataset, which is a representative sample of the United States population that represents the cross-sectional population of the United States after survey weighting. The percentage on Table 1 is not “percent of the total” per se, but the survey-weighted percentage that represents the US population. We had a footnote 2 on Table 1 that states “Results were obtained after adjusting for sample weights and complex study design” to explain this. We did not aim to only represent a certain ethnic/racial population, we wanted to represent the population of the US through this sample.
Comment 3: The study completely lacks investigation on the cultural level of the people: instruction level, cultural habits that play a crucial importance in defining drinking and eating habits too. This point must be added, and its weight well specified in the study.
Response 3: Thank you for the comment, the data on cultural level of people is certainly lacking in the NHANES dataset because NHANES did not collect such variables. We will include this as a limitation in our discussion in lines 396-400: “Finally, the potential cultural impact of diet and alcohol behavior were not considered because of the limitations of the NHANES dataset, and future prospective studies collecting cultural variables would be beneficial in elucidating the influence of so-cial/cultural norms on both alcohol use and dietary quality.”
Comment 4: The poverty to income ratio must be well specified in the text. This reviewer did not well understand the meaning of the numbers of the different groups, nor how these groups were decided and adopted.
Response 4: We expanded the section on PIR in lines 140-145 of revised manuscript: “PIR is defined as the total household income divided by the poverty threshold for household size as defined by the US census bureau. PIR serves as a standardized measure of socioeconomic status that allows for comparison of individual income across geography and time within the large area of the United States and is classified into 5 categories of increasing income: <1.00 (below poverty), 1.00-1.99, 2.00-2.99, 3.00-3.99, and ≥ 4.” We added citations in the revised manuscript from the census bureau which uses PIR and a book title that initially introduced the concept of PIR. For your convenience, the references are below.
Added citations:
United States Census Bureau. Poverty Thresholds by Size of Family and Number of Children. Availabe online: https://www.census.gov/data/tables/time-series/demo/income-poverty/historical-poverty-thresholds.html
Citro, C.F.; Michael, R.T.; Panel on Poverty and Family Assistance (United States). Measuring poverty : a new approach; National Academy Press: Washington, D.C., 1995; pp. xviii, 501 p.
Comment 5: No other exclusion criteria than alcohol abuse? It seems very narrow to limit and define the investigated set.
Response 5: We did not set alcohol abuse as an exclusion criteria. In fact, we included all alcohol drinkers to better understand the differences in diet among different types of alcohol drinkers.
Comment 6: The authors defined an only-bear and an only-wine groups. Indeed, USA is a country where the daily use of wine is not present in the population habits as it would be for Mediterranean (Spanish, Italian, French) population. The reason is a cultural, agricultural and hence economic weight. Which the reason why the authors choose only-wine as one of the groups for their study? In this reviewer’s opinion this choice is not as obvious as it would be in Italy or French. This point would have been defined and discussed in the manuscript.
Response 6: Thank you for the comment, the aim of our study was to investigate diet quality differences of individuals according to the alcoholic beverage consumed in the timeframe of the diet consumption (24 hour dietary recall survey). Beer-only, wine-only and liquor/cocktail-only consumers were identified based on What We Eat In America (WWEIA) Category number, which only distinguishes between these 3 types of alcohol. Individuals who consumed 2 or more alcoholic beverage types within the 24 hour survey period were defined as a fourth type of alcohol consumer: multiple-types. We added a reference (shown below) and amended the method section to be more concise on this point in lines 107-109: Using the "What We Eat in America" Food Categories in the 24-hour dietary recall survey, we classified alcohol drinkers into 4 groups: beer-only, wine-only, liquor/cocktail-only, and multiple types (if consumed ≥2 types of alcoholic beverages).
Added citations:
What We Eat in America Food Categories for use with WWEIA, NHANES 2017 - March 2020 Prepandemic. Available online: https://www.ars.usda.gov/northeast-area/beltsville-md-bhnrc/beltsville-human-nutrition-research-center/food-surveys-research-group/docs/dmr-food-categories/ (accessed on Oct 09, 2024).
Comment 7: Based on the above observation, would the authors think possible to apply their study to other countries different from USA and what do the authors expect?
Response 7: This is an essential question that, as your previous comment on cultural aspects pointed out, is difficult to ascertain. On one hand, it is almost impossible to say that this study will apply perfectly to countries outside the USA, because it was designed and studied in a US population using the NHANES database. By that fact, any application to other countries would be extrapolation. On the other hand, cultural pairings between alcohol and food, and the societal/cultural norms around alcohol, have been converging due to an extended period of globalization and technology. For example, whereas 50 years ago it would be difficult to imagine American fast food and alcoholism to be the drivers of health issues in many parts of the developing world, both fast food and alcoholism are now pervasive. This is one of the impetuses that led to our study.
Comment 8: The limitations of the study are vague and difficult to understand in their practical application.
Response 8: We added cultural aspect to our limitations as you suggested in comment 3 and added that a practical application of this limitation is that future studies should be longitudinal and/or prospective to make sure cultural aspects are included in prospective data collection.

Reviewer 2 Report
Comments and Suggestions for Authors
The manuscript entitled “Convergence of Alcohol Consumption and Dietary Quality in US Adults Who Currently Drink Alcohol - An Analysis of Two Core Risk Factors of Liver Disease” presents interesting issue but there are serious problems.
Major:
(1) The major limitation of the presented manuscript is associated with the fact that Authors studied only current alcohol drinkers with no control group of respondents not being a current alcohol drinkers. Taking this into account, Authors can not conclude about “real” association/ convergence between drinking alcohol and diet quality, as they only compared respondents drinking various types of alcohol beverages. As Authors used the NHANES data, they should include the non-drinkers population in order to have a control group for their study.
(2) The study was conducted before and during the COVID-19 pandemic, as data from the period of 2017-2020 were included. The first problem is associated with the fact that the COVID-19 pandemic significantly influenced alcohol consumption, so data for 2017 and for 2020 are incomparable. Authors should compare the data gathered before COVID-19 pandemic (2017-2019) and during the COVID-19 pandemic (2019-2020) in order to define if their population was influenced by this factor.
(3) The other problem associated with the COVID-19 pandemic results from a serious changes of diet quality (and alcohol consumption as well) after the pandemic. As a result, data gathered before pandemic are now out of date and can not be indicated as current and valid for the nowadays.
(4) Authors did not take into account that the consumption of specific alcohol beverages is associated with specific dietary habits (within a meal with the alcoholic beverage), so it is not known if it influenced.
(5) It seems that Authors interpreted drinking alcohol at least once during a year as being alcohol drinker, but it is not clearly stated (“Current alcohol drinkers were defined by NHANES Alcohol Use Questionnaire (ALQ) as individuals who had consumed alcohol within the past 12 months”). If so, it may have contributed to serious inaccuracy, as individual who had one drink during a year can not be assessed within the same group with other drinking alcohol each day. It should be explained clearly.
(6) Authors referred HEI-2015, but not the current HEI-2020.
(7) Authors indicated that they assessed food security, but they did not explain how did they do it. The only information was as follows: “ food security was assessed using a 10-item questionnaire evaluating the level of food security over the past 12 months. The responses were totaled and classified into one of four categories: full food security, marginal food security, low food security, and very low food security.” – Authors should specify the applied questionnaire, responses allowed, if the questionnaire was validated or not, etc.
Author Response
Comment 1: The major limitation of the presented manuscript is associated with the fact that Authors studied only current alcohol drinkers with no control group of respondents not being a current alcohol drinkers. Taking this into account, Authors can not conclude about “real” association/ convergence between drinking alcohol and diet quality, as they only compared respondents drinking various types of alcohol beverages. As Authors used the NHANES data, they should include the non-drinkers population in order to have a control group for their study.
Response 1: Thank you for the comment. The aim of our study was to assess the association between diet quality and alcoholic beverage type choice, rather than the association between diet quality and alcohol intake. You are absolutely right about our inclusion criteria, but the reason why we chose to study the association between alcoholic beverage type and diet quality among all drinkers is because this is where the knowledge gap is. In contrast, we detailed in our discussion section 4.3 that prior studies did show worse diet quality with increasing quartiles of alcohol use or episodes of binge drinking; those studies already established that alcohol drinkers had lower diet quality compared to alcohol abstainers.
Comment 2: The study was conducted before and during the COVID-19 pandemic, as data from the period of 2017-2020 were included. The first problem is associated with the fact that the COVID-19 pandemic significantly influenced alcohol consumption, so data for 2017 and for 2020 are incomparable. Authors should compare the data gathered before COVID-19 pandemic (2017-2019) and during the COVID-19 pandemic (2019-2020) in order to define if their population was influenced by this factor.
Response 2: The study did not include the post-pandemic 2020 period, it only included 2017 up until 2020 prior to the pandemic up until March 2020. Up until March 2020, the US (where the NHANES dataset we used is administered) was still pre-pandemic and there were no lockdowns. During the pandemic, trained interviewers were not able to obtain information at household visits. In our method section, we specified that our dataset is pre-pandemic “We utilized the pre-pandemic 2017-2020 cycle of the National Health and Nutrition Examination Survey (NHANES).” We added a citation in the manuscript, we included it below for convenience.
Added citations:
Centers for Disease Control and Prevention (CDC); National Center for Health Statistics (NCHS). NHANES Survey. Methods and Analytic Guidelines. Available online https://www.cdc.gov/nchs/nhanes/index.htm. (accessed on March 5, 2023).
Comment 3: The other problem associated with the COVID-19 pandemic results from a serious changes of diet quality (and alcohol consumption as well) after the pandemic. As a result, data gathered before pandemic are now out of date and can not be indicated as current and valid for the nowadays.
Response 3: The NHANES cycle from 2017 to March 2020 is the most recent available dataset representing the US population that can be used to investigate our study purpose. We agree that COVID 19 may have had uncertain effects on both diet quality and alcohol. We added to discussion section in lines 390-392 : “The COVID-19 pandemic occurred after the study period, and we recognize both dietary and alcohol behaviors may have changed after the pandemic in uncertain ways that our study was not designed to detect.”
Comment 4: Authors did not take into account that the consumption of specific alcohol beverages is associated with specific dietary habits (within a meal with the alcoholic beverage), so it is not known if it influenced.
Response 4: The diet quality (including components of the healthy eating index and macronutrient consumption) was, in fact, collected in the same 24-hour dietary recall questionnaire as the specific alcoholic beverage. This puts the meal and the alcohol consumption in the same 24-hour window of consumption. You are right that within that 24-hour period, we cannot determine whether the alcoholic beverage was consumed in the same exact meal, and we had added to our limitations suggesting future prospective studies would be more equipped with assessing the specific meal taken with a specific alcoholic beverage (which is often a cultural behavior).
Comment 5: It seems that Authors interpreted drinking alcohol at least once during a year as being alcohol drinker, but it is not clearly stated (“Current alcohol drinkers were defined by NHANES Alcohol Use Questionnaire (ALQ) as individuals who had consumed alcohol within the past 12 months”). If so, it may have contributed to serious inaccuracy, as individual who had one drink during a year can not be assessed within the same group with other drinking alcohol each day. It should be explained clearly.
Response 5: Thank you for the suggestion. We reworded our methods in lines 102-104 to be more accurate on our inclusion criteria to avoid misunderstanding: “We included participants who were ≥20 years-old, identified as current alcohol drinkers by the 12-month Alcohol Use Questionnaire (ALQ), and consumed alcohol within a completed 24-hour dietary recall.” We recognize that our methods are quite strict in defining alcohol use because we wanted consistency between the different alcohol use surveys that are embedded within NHANES dataset. We wanted both the ALQ questionnaire to identify the subject as an alcohol user within the past 12 months, AND that they drank within the 24-hour recall survey which is used to identify the alcoholic-beverage type. The revision to our methods hopefully clarifies this point.
To your second point, which astutely points out regular/heavy drinking is clearly different from rare episodic drinking that may be both positive for the ALQ, we accounted for the level of chronic alcohol use using the ALQ questionnaire to categorize regular/heavy drinkers. We did this in case there was a difference in diet quality between these groups. We then adjusted for heavy drinking (as defined by NIAAA) in our multiple linear regression models. We did not find that current heavy drinkers in our population had a difference in diet quality compared to nonheavy drinkers, but remote heavy drinkers seemed to have worse diet quality (which is shown in Supplementary Table 2). Because of this finding, we further performed sensitivity analysis to exclude those with remote heavy drinking in case this would affect our results, but we found the association between beer-only drinking and poor dietary quality remained (shown in Supplementary Table 3)
Comment 6: Authors referred HEI-2015, but not the current HEI-2020.
Response 6: The study design and data collection for NHANES cycle 2017 to March 2020 was administered based on the HEI-2015. HEI-2020 was applied after the NHANES cycle 08/2021to 08/2023. However, the dietary dataset from the cycle 08/2021 to 08/2023 has yet not been released, which means our dataset is the most recent available dataset for our research question.
Comment 7: Authors indicated that they assessed food security, but they did not explain how did they do it. The only information was as follows: “ food security was assessed using a 10-item questionnaire evaluating the level of food security over the past 12 months. The responses were totaled and classified into one of four categories: full food security, marginal food security, low food security, and very low food security.” – Authors should specify the applied questionnaire, responses allowed, if the questionnaire was validated or not, etc.
Response 7: We added a citation from the US Department of Agriculture which details food security definitions (shown below for convenience). The reason we did not expand heavily upon food security in the methods is because we ultimately did not adjust for it in the multivariate regression.
We originally wanted to include measure of food security because we thought it might impact diet quality. However, we performed correlation analysis and found poverty income ratio (PIR) and food security were highly correlated. Thus, we included PIR, but not food security, in multivariate regression. We wanted to still present the food security data in Table 3 as we are investigating nutrition as the outcome.
Added citations:
U.S. Department of Agriculture Economic Research Service. Definitions of Food Security. Availabe online: https://www.ers.usda.gov/topics/food-nutrition-assistance/food-security-in-the-u-s/definitions-of-food-security/ (accessed on Oct 10, 2024).

Round 2
Reviewer 1 Report
Comments and Suggestions for Authors
I think the revised manuscript now suitable for publication in Nutrients journal
Comments on the Quality of English Languagemoderate editing requested
Author Response
Comment 1: moderate editing requested
Response 1: Thank you for your suggestion. We edited the entire manuscript to correct grammatical errors and improve clarity, including but not limited to the following:
- Ensuring correct tense usage, applying past tense for past research and present tense for generalizations or current theories,
- Eliminating run-on sentences throughout the manuscript,
- Verifying proper punctuation usage (e.g., commas),
- Choosing concise language wherever possible,
- Preferring active voice language wherever appropriate, and
- Correcting any other grammatical, syntactical, or diction errors found.
Reviewer 2 Report
Comments and Suggestions for Authors
The manuscript entitled “Convergence of Alcohol Consumption and Dietary Quality in US Adults Who Currently Drink Alcohol - An Analysis of Two Core Risk Factors of Liver Disease” presents interesting issue but there are serious problems.
Major:
(1) The major limitation of the presented manuscript is associated with the fact that Authors studied only current alcohol drinkers with no control group of respondents not being a current alcohol drinkers. Taking this into account, Authors can not conclude about “real” association/ convergence between drinking alcohol and diet quality, as they only compared respondents drinking various types of alcohol beverages. As Authors used the NHANES data, they should include the non-drinkers population in order to have a control group for their study. If Authors did not, they should explain it in their manuscript and justify it presenting research gap in the Introduction Section.
(2) Authors did not take into account that the consumption of specific alcohol beverages is associated with specific dietary habits (within a meal with the alcoholic beverage), so it is not known if it influenced. Authors should mention it in their Discussion Section, as they state that e.g. “beer consumption is associated with lower relative dietary quality compared to wine consumption” (while beer is drunk what are typical products consumed? while wine is drunk what are typical products consumed? Is there any potential association? For sure it is)
(3) It seems that Authors interpreted drinking alcohol at least once during a year as being alcohol drinker, but it is not clearly stated (“Current alcohol drinkers were defined by NHANES Alcohol Use Questionnaire (ALQ) as individuals who had consumed alcohol within the past 12 months”). If so, it may have contributed to serious inaccuracy, as individual who had one drink during a year can not be assessed within the same group with other drinking alcohol each day. It should be explained clearly.
(4) Authors referred HEI-2015, but not the current HEI-2020. Authors should get familiar with HEI-2020 – it does not differ from HEI-2015, but it is recommended to refer this HEI as the most current.
(5) In my opinion Authors should remove the part associated with food security, as it is not associated with general part of the manuscript
Author Response
Comment 1: The major limitation of the presented manuscript is associated with the fact that Authors studied only current alcohol drinkers with no control group of respondents not being a current alcohol drinkers. Taking this into account, Authors can not conclude about “real” association/ convergence between drinking alcohol and diet quality, as they only compared respondents drinking various types of alcohol beverages. As Authors used the NHANES data, they should include the non-drinkers population in order to have a control group for their study. If Authors did not, they should explain it in their manuscript and justify it presenting research gap in the Introduction Section.
Response 1: Thank you for the specific suggestion. We did not include non-drinkers because nondrinkers were shown to have better diet quality than alcohol drinkers in a dose dependent association in prior NHANES cohorts, we had also previously included in our discussion in section 4.3 under Associations of Alcohol Consumption Pattern with Dietary Quality a detailed literature review of this subject.
The hypothesis of this study is that diet quality may differ among different types of alcoholic beverages consumers. We specifically focused on examining the correlation between alcohol type and diet quality independent of the amount of ethanol consumed. Certainly the amount of alcohol consumption matters, which is why we took measures to account for amount of alcohol use (please see response to your comment 3). The amounts of alcohol use were categorized into light, heavy, and remote history of heavy drinking, and this variable was considered in the multivariate regression models when evaluating the association between diet quality and alcohol type consumption.
Since prior studies have already had assessed alcohol consumption patterns (including nondrinkers) and diet quality, we added/amended our Introduction section in the new revised manuscript in lines 62-66 for explanation:
“Prior research had demonstrated a dose-dependent worsening of diet quality with increasing alcohol consumption in a cohort that included non-drinkers, but did not take into account alcoholic beverage type. To this day, the specific relationship between the type of alcoholic beverage consumed (e.g. wine, beer, liquor) and dietary quality remains a research gap.”
Comment 2: Authors did not take into account that the consumption of specific alcohol beverages is associated with specific dietary habits (within a meal with the alcoholic beverage), so it is not known if it influenced. Authors should mention it in their Discussion Section, as they state that e.g. “beer consumption is associated with lower relative dietary quality compared to wine consumption” (while beer is drunk what are typical products consumed? while wine is drunk what are typical products consumed? Is there any potential association? For sure it is)
Response 2: Thank you for specifying your suggestion and we agree that there is a potential association, but were surprised the literature was sparse on the cultural/social science investigation of alcohol and food pairings. We did add one citation (added below for convenience) which was an unadjusted descriptive study on “situated drinking” that had data on the context of different alcohol beverage type with meals. We amended our limitation on this subject in lines 399-404:
“Finally, the potential cultural impact of diet and alcohol behavior were not considered because of the limitations of the NHANES dataset in identifying cultural variables such as the pairing of specific alcoholic beverage types to food in the same meal, and future prospective studies collecting data on cultural context would be beneficial in elucidating the influence of social/cultural norms on both alcohol use and dietary quality.”
We also added in lines 291-299 to specifically propose a hypothesis for our findings:
“One hypothesis for the association we found is the context of consumption of alcohol may influence dietary behavior and vice versa. For instance, a full course meal that contains fruits and vegetables with high diet quality often accompanies wine or spirit-based drinks, and high sodium or fried foods with low diet quality may demand carbonated alcoholic beverages such as beer. This is an area that remains uninvestigated, but a descriptive study set in Great Britain reported wine is more often paired with a formal meal (45.8%, vs. 25.6% for beer), whereas beer is more often paired with a snack (33.2%, vs. 25.6% for wine). However, due to the limitation of the 24-hour dietary recall instrument, we could not localize the alcohol consumption to each specific meal.”
Added citation:
Warde, A.; Sasso, A.; Holmes, J.; Hernandez Alava, M.; Stevely, A.K.; Meier, P.S. Situated drinking: The association between eating and alcohol consumption in Great Britain. Nordisk Alkohol Nark 2023, 40, 301-318, doi:10.1177/14550725231157222.
Comment 3: It seems that Authors interpreted drinking alcohol at least once during a year as being alcohol drinker, but it is not clearly stated (“Current alcohol drinkers were defined by NHANES Alcohol Use Questionnaire (ALQ) as individuals who had consumed alcohol within the past 12 months”). If so, it may have contributed to serious inaccuracy, as individual who had one drink during a year can not be assessed within the same group with other drinking alcohol each day. It should be explained clearly.
Response 3: Thank you for the suggestion again. In our last revision, we amended our methods in lines 104-106 to be more accurate on our inclusion criteria to avoid misunderstanding:
“We included participants who were ≥20 years-old, identified as current alcohol drinkers by the 12-month Alcohol Use Questionnaire (ALQ), and consumed alcohol within a completed 24-hour dietary recall.” This statement clearly states that current alcohol drinker means both alcohol use within past 12 months AND documented alcohol use in the 24 hour dietary recall survey, and this is because we wanted consistency between the different alcohol use surveys that are embedded within NHANES dataset. Hopefully we have clarified this point.
We completely recognize your point that individuals who had one drink per year cannot be viewed in the same way as those drinking alcohol each day (even though both would be positive for ALQ), which is why we also collected data on the level of alcohol use chronically over the course of the year using ALQ121 and ALQ130. We revised the method section in lines 148-152 to clarify that we used these two instruments to calculate the level of drinking:
“We calculated the quantity of alcohol consumption per week using two questions within the NHANES Alcohol Use Questionnaire (ALQ) to identify heavy drinking: ALQ121: "How often did you drink any type of alcoholic beverage during the past 12 months?" and ALQ130: "On average, how many drinks did you have on those days you drank alcoholic beverages?"
We did this in case there was a difference in diet quality between heavy drinkers and non-heavy drinkers. We then adjusted for heavy drinking (as defined by NIAAA detailed in our methods section in lines 146-148) in our multiple linear regression models. We did not find that current heavy drinkers in our population had a difference in diet quality compared to nonheavy drinkers, but remote heavy drinkers seemed to have worse diet quality (which is shown in Supplementary Table 2). Due to this observation of poor diet in remote heavy drinkers, we further performed sensitivity analysis (methods described in lines 193-197) to exclude those with remote heavy drinking in case this would affect our results, but we ultimately found the association between beer-only drinking and poor dietary quality remained and our conclusion did not change (shown in Supplementary Table 3)
Comment 4: Authors referred HEI-2015, but not the current HEI-2020. Authors should get familiar with HEI-2020 – it does not differ from HEI-2015, but it is recommended to refer this HEI as the most current.
Response 4: Thank you, we will definitively use HEI-2020 for future publications. On a practical level, the SAS code used for calculating HEI-2020 from the NHANES dataset is not yet available/released. Please see here for reference, where the National Cancer Institute (NCI) notes that “SAS code for the HEI-Toddlers-2020 and the HEI-2020 is coming soon”: https://epi.grants.cancer.gov/hei/sas-code.html
We agree that after much research on the subject, it appears that HEI-2015 and HEI-2020 do not seem different (at least in those age 2 or older). However, the specific dietary recommendations for each nutrient intake that are used to score each HEI component may be different between HEI-2015 and HEI-2020, because HEI-2015 is based off of the 2015-2020 Dietary Guidelines of America (DGA) whereas the HEI-2020 uses the 2020-2025 DGA. The priorities of 2020-2025 DGA may be different than that of 2015-2020 DGA, which may alter HEI in ways we cannot predict. Without the SAS codes, we cannot be certain our HEI-2015 is equivalent to HEI-2020, even though the scoring system appear to be the same.
Comment 5: In my opinion Authors should remove the part associated with food security, as it is not associated with general part of the manuscript
Response 5: Thank you for the suggestion, we removed it from the manuscript.